# Navigating parenting challenges: A qualitative study of communication experiences among young Chinese couples facing breast cancer while raising underage children

Yingchun Li[1,2], Meichan Chong[1*], Pinglei Chui[1], Lin Mo[3], Liande Tao[2], Yuman Yuan[4], Qiaoli Zhong[4], Maoting Tang[5], Yanjia Liu[1]

**1** Department of Nursing Science, University of Malaya, Kuala Lumpur, Malaysia, **2** Nursing Department, Yibin Second People's Hospital, Yibin, China, **3** Affiliated Children's Hospital of Chongqing Medical University, Chongqing, China, **4** Breast and Thyroid Department, Yibin Second People's Hospital, Yibin, China, **5** Department of Pediatric Intensive Care Unit Nursing, West China Second University Hospital, Chengdu, China

* mcchong@um.edu.my

## Abstract

This qualitative study aimed to gain an in-depth understanding of the communication experiences of young couples with breast cancer as they navigate the challenges of raising underage children. Semi-structured interviews were conducted with 13 couples with breast cancer recruited through purposive sampling in the inpatient ward of a tertiary hospital in Sichuan Province, China. The interviews, conducted face-to-face between May 2024 and June 2024, were audio-recorded, transcribed verbatim, and analyzed using thematic analysis. This study was guided by the Consolidated Criteria for Reporting Qualitative Research (COREQ) checklist. Two researchers independently coded the data using NVivo 12.0, developing major themes and subthemes through an inductive process and constant comparison. The analysis revealed two main themes, each encompassing three to four subthemes. The first theme, dyadic coparenting through constructive communication: a shared journey of navigating parenting challenges, which includes open communication of parenting experiences, emotional co-regulation in parenting, and positive communication of parenting plans; The second theme, negative co-parenting communication: protect parenting emotions or not provide emotional support, which includes "instrumental support" and emotional avoidance, avoid communication to prevent emotional fluctuations, communication concealment to avoid increasing parenting pressure, and invalid communication without providing parenting emotional support. These findings highlight the diverse communication strategies employed by young couples with breast cancer in parenting their underage children. The study underscores the need for healthcare professionals to develop targeted interventions to enhance communication between young breast cancer couples, with the aim of improving their ability to co-parent effectively during this challenging time.

**Data availability statement:** All relevant data are publicly available from the figshare repository (https://doi.org/10.6084/m9.figshare.30050026).

**Funding:** The author(s) received no specific funding for this work.

**Competing interests:** The authors have declared that no competing interests exist.

## Introduction

According to the latest cancer statistics released by the National Cancer Registry [1], breast cancer ranks first in female cancer incidence in China, which accounting for 16.7% of all new cases of cancer in women and ranks fourth in female cancer mortality. As breast cancer increasingly affects younger individuals and survival rates improve, a growing number of breast cancer patients find themselves parenting underage children (children under the age of 18) [2]. Young cancer patients pose a serious challenge to families, as patients must strike a balance between managing the disease and caring for underage children [3]. The issue of "parenting concerns" or "child-bearing concerns" was first highlighted by Hymovich that the parenting concerns of cancer patients are stressful emotions caused by the inability of patients to balance raising children with disease diagnosis and treatment [4].

Parenting Concern (PC) refers to the anxiety emotions generated by cancer patients due to their concern about their children's parenting issues. The PC of cancer patients includes not only worries about the impact of their illness on their children but also apprehensions regarding their spouse's ability to effectively parent and care for their children [5]. Compared with other tumor, breast cancer affects patients' parenting function in a unique way. On the one hand, breast surgery can cause damage to the patient's body image, which can restrict patients from having physical contact with their children [6]. They are concerned that their incomplete image will have a negative emotional impact on their children. In addition, breast surgery may also cause interference to the patient's breastfeeding ability [6]. On the other hand, for breast cancer patients undergoing axillary lymph node dissection, in order to prevent the occurrence of lymphedema, they are usually restricted from heavy physical activities such as holding children, which hinders the close contact between breast cancer patients and their children, which will also increase the concern of breast cancer patients about parent-child relationship [7]. These conditions are unique characteristics of breast cancer, which makes patients unable to perform the role of mother as before diagnosis, which can lead to patients' guilt and feeling that they are not a competent mother [8]. It can be seen that breast cancer diagnosis will threaten the self-efficacy, role and identity of breast cancer patients, which is easy to cause parenting concerns [9]. Research has shown that young cancer patients and their spouses who raise underage children generally have a certain level of PC [10]. In China, due to the long-term influence of Confucian culture that prioritizes family, parents value taking care of their children personally and concern that their children cannot survive or grow up without them [11,12]. They regard raising children as their duty and pay more attention to their growth and development. When patients cannot balance the roles of parents and patients, and cannot fulfill their parenting responsibilities well, they will be more concerned about the impact of their illness on their children [5]. PC persists throughout the patient's diagnosis and treatment journey, significantly influencing their physical and mental health, as well as impacting decisions related to their medical treatment7. Among cancer patients with parenting concerns, 24% to 71% exhibit varying degrees of anxiety and depression caused by parenting issues [13]. Depression can exacerbate the psychological distress of

cancer patients, reduce their participation in medical care, and have a negative impact on treatment outcomes of up to 20%, leading to longer hospital stays [14].

Spouses are the primary caregivers and emotional supporters of cancer patients. Effective communication between cancer patients and their spouses can help enhance their emotional support and reduce psychological distress [15]. Communication mode refers to the way couples perceive themselves to communicate and interact with each other [11]. Up to date, studies on the couple communication mainly focused on diseases such as colorectal cancer [16,17], prostate cancer [18], gastric cancer [19], and glioma [17]. With regard to the communication between breast cancer patients and their spouses, existing researches shown that the communication is mainly manifested in role conversion, symptom relief, social communication and disease adaptation [20], and less communicate about anxiety, depression, parenting and other topics. Fried et al. [21]further pointed out that the desire for disease communication between breast cancer patients and their spouses has not been met, and they believe that poor disease communication is an important source of marital conflicts. Multiple studies have shown that couples with limited communication during disease treatment are more likely to experience psychological distress, lower marital satisfaction, and more family conflicts [16,22]. Xiao Ting's study shown that there was a common problem of couple communication between breast cancer patients and their spouses [23]. Breast cancer patients and their spouses tend to avoid topics such as the patient's disease progress, disease prognosis, death and fear and one partner often lacks the correct response from the other when expressing emotions, which further increasing the psychological burden of both breast cancer patients and their spouses [23].

To sum up, there is a certain degree of couple communication problems between breast cancer patients and their spouses, which can lead to negative impact on the health outcomes of breast cancer patients and their spouses. Regarding to parenting underage children, most existing studies focus on a single perspective of cancer patients. Some cross-sectional studies aim to investigate the PC levels of cancer patients [3] and some qualitative researches aim to explore the experience of cancer patients raising underage children [5], few studies consider the influence of spouses on patients raising underage children and there is currently no research exploring the communication between breast cancer patients and their spouses in raising underage children.

Given that communication plays an important role in coping with psychological stress in cancer patients and their spouses, and due to the unique cultural background in China, the communication experience in parenting underage children among young breast cancer patients and their spouses in China warrants further exploration. The purpose of the study is to explore the parenting-related communication experience between breast cancer patients and their spouses, laying a foundation for the development of intervention strategies for breast cancer patients to navigate parenting challenges.

## Methods

### Design and setting

A qualitative design was used to explore the communication experience in parenting underage children among young breast cancer patients and their spouses in China. The study was conducted at Yibin Second People's Hospital. Reporting was performed in accordance with the Consolidated Criteria for Reporting Qualitative Research (COREQ) reporting guidelines [24].

### Participants and sampling

The participants were recruited using purposive sampling with maximum variation to ensure sample diversity in terms of education level and stage of cancer from 09/05/2024–30/06/2024. Participant recruitment was conducted after the patient is admitted. Reception nurse (ZQL) screened potential participants (including young breast cancer patients and their spouse) according to inclusion and exclusion criterias and inquired about their willingness to participate, only those both patients and their spouses were agreed to participate would be recruited. If the young breast cancer patients and their spouses are willing to participate, the reception nurse notify the interviewer (YYM) to conduct the interview. During the

recruitment process, participants were recruited based on cancer stage (I–III), education level (high school or below/ college/ undergraduate/postgraduate), parental status (number and age of children), and family residence (city/town). The research team reviewed the distribution of included samples on a weekly basis and dynamically adjusted subsequent recruitment priorities until with no new characteristic patterns emerging.

The potential participants were patients who met the following inclusion criteria: (1) age 20–44 years old; (2) pathological diagnosis of breast cancer, and knowing their condition; (3) currently raising at least one child under the age of 18; (4) physically able to participate; and (5) willing to provide informed consent to participate in this study. The inclusion criteria for spouses were: (1) those who are aware of patient's diagnosis; (2) informed consent and voluntary participation in this study. Patients or their spouses with cognitive or mental disorders that inhibit cooperation, with serious comorbidities were excluded. The sample size was determined by data saturation, which was assessed according to the method from Guest [25]. In this study, no new themes emerged after the 11th interview. According to the method proposed by Guest, a total of 21 new themes were generated from interviews 1–9, while only one new theme was identified from interviews 10–11. This represented an incremental yield of 4.7% (1/21), which fell below the predetermined saturation threshold of 5%, indicating that saturation was achieved. Two additional interviews were then conducted, and no new themes were observed. By the 13th interview, the principle of maximum participant variation was satisfied. Therefore, a final sample of 13 participants was included in this study.

## Data collection

The qualitative interview forms that include family members can mainly be divided into individual interviews, dyadic/group interviews and a combination of individual and dyadic/group interviews [26]. This study aimed to obtain information on parenting-related communication experience between patients and spouses. Considering the advantages and disadvantages of individual interviews and dyadic/group interviews, individual interviews can ensure that participants can freely express their personal views and opinions without interference from others, therefore, individual interviews were adopted. In this study, breast cancer patients and their spouses were interviewed separately.

Literature review and group discussion were conducted to develop an interview outline [27]. Prior to the formal interview, three couples of young breast cancer patients were selected for pre interviews. Several modifications on inquiry method were made according to the interviews to form the formal interview outline. For example, we modified patient's interview outline"After falling ill, how does your husband communicate with you about their thoughts on child parenting?" to "Would you tell your husband about your concerns about your child? Why?". The semi-structured interview outline for patient is like this: (1) Are you concerned about your child since falling ill? What are you concern about? (2) Would you tell your husband about your concerns about your child? Why? (3) What was your husband's response after you told him about your concerns about your child? Did your husband communicate with you? (4) Did your husband's responses affect you? How did that make you feel? The semi-structured interview outline for spouse is like this: (1) Are you concerned about your child since your wife fell ill? What are you concern about? (2) Would you tell your wife about your concerns about your child? Why? (3) What was your wife's response after you told her about your concerns about the child? Did your wife communicate with you? (4) Did your wife's responses affect you? How did that make you feel?

The interview was conducted in the conversation room of the breast department. Face-to-face interviews were individually conducted by a researcher (YYM) with extensive experience in qualitative interviewing. The researcher is the head nurse with master's degree (46 years old, associate chief nurse, female), and had been working in breast department for over 10 years. In the process of this research, the interviewer played the dual roles of "insider" and "outsider". As "insiders," the interviewer was head nurse of her respective breast department. During her clinical work, she had previously had the opportunity to communicate with patients with breast cancer and had formed a good nurse-patient relationship. As "outsiders," the interviewer only performed nursing and management work and had rarely discussed issues related to PC with patients to avoid preconceived experiences or impressions affecting the results.

Before the interview, the interviewer (YYM) introduced the research purpose to the participants, ensured strict confidentiality of the interview content. Interviews for patients and spouses were conducted at the interviewee's free time, when feeling up to the effort of answering questions. After obtaining the written informed consent of each interviewee, the interview was recorded. Questions were asked in an open and non-inductive manner, allowing the interviewee to fully express his or her true thoughts and feelings. the interviewer paid attention to the body language of the interviewee, such as facial expressions and movements, and kept records throughout the interview. During the interview process, techniques such as questioning, retelling and clarifying were used to avoid giving any hints. Each interviewee was interviewed for 30~40 minutes.

## Data analysis

The interviewer and one researcher (ZQL) listened to the audio recordings repeatedly and transcribed them into text within 24 hours of the interview to ensure transcript accuracy. The interview transcript analysis was performed by two researchers (TMT and LYC) independently using NVivo12 software. Transcripts were analyzed line-by-line and coded based on the research questions. To protect the privacy of the participants, P was used to replace the patient, and S was used to replace the spouse.

The dyadic interview analysis method [28] was used to analyze the data. This method involves following the initial steps of thematic analysis, such as reading and re-reading the interviews, making notes, and coding the data to identify units of meaning and their corresponding titles [29]. This process was repeated for each participant's interview, and all 26 interviews were analyzed. The analysis, conducted by TMT and LYC, was conducted on both individual and dyadic levels to examine the data from different perspectives [29].

Two researchers (TMT and LYC) independently performed open coding on the first three interview transcripts, generating preliminary codes based on meaningful units related to parenting communication. These initial codes were then compared and discussed to develop a preliminary coding framework. This framework was subsequently applied to the remaining transcripts, with ongoing refinement as new concepts emerged. Throughout the analysis, codes were constantly compared, merged, split, or discarded through iterative discussions between the two coders. Memos were maintained to document coding decisions and track the evolution of themes, ensuring an auditable trail of analytical choices [30]. At the individual level, researchers (TMT and LYC) independently code the interview data of patient and spouse separately, and generate two sets of initial codes, each participant's quotes were analyzed in an in-depth fashion, with content, structure, linguistics, grammar, and metaphors being taken into consideration [30]. At the dyadic level, the focus was on examining the contrasts and overlaps between breast cancer patients and their spouses' quotes [28]. The themes were identified on the basis of the content and structure of the quotes specifically related to the parenting communication. To ensure coding reliability and minimize individual bias, discrepancies between the two researchers (TMT and LYC) were systematically addressed through regular consensus meetings. When coding disagreements arose, the researchers (TMT and LYC) first revisited the relevant transcript segments together and discussed their respective interpretations, referring back to the research questions and the evolving coding framework. If consensus could not be reached through discussion, a third researcher (CMC) with extensive qualitative research experience was consulted to provide an independent perspective. The third researcher (CMC) reviewed the disputed codes and corresponding data, facilitating a final decision through team discussion until full agreement was achieved. This iterative process of discussion and verification continued throughout the analysis phase, and all coding decisions were documented in an audit trail. Finally, all research team members reviewed and unanimously approved the final themes in a concluding discussion meeting [31].

## Trustworthiness

The trustworthiness of this qualitative study was ensured by maintaining credibility, transferability, dependability, and conformability [32]. To ensure the credibility of this qualitative analysis, we adopted persistent observation, i.e., the

researcher paying attention to the feelings or emotions of the interviewees [33], to identify the most relevant information related to the participants' experiences. We also routinely read and reread the data, analyzed them, and revised the extracted concepts until the final abstracting provided an in-depth view of the participants' experiences [34]. We attempted to describe the results thoroughly and clearly to enhance transferability [34]. To ensure dependability, we reported our process in detail [35]. In addition, investigator triangulation was adopted by using two researchers to make coding, analysis, and interpretation decisions to develop more objective results and avoid misinterpretation or subjective imagination [36]. Apart from this, to mitigate potential bias arising from the interviewer's dual role as both an "insider" and "outsider," several strategies were implemented. First, the semi-structured interview guide was collaboratively developed and reviewed by external experts to ensure neutrality and openness in questioning. Second, during interviews, the interviewer adopted an "empathic neutrality" approach, balancing rapport-building with professional detachment, and used probing techniques to explore participants' unique experiences rather than relying on prior familiarity. Third, investigator triangulation was employed, with at least two researchers conducting independent data analysis followed by team discussions to challenge interpretations and minimize individual bias. Finally, member checking with a subset of participants verified the accuracy and resonance of the findings. These measures collectively enhanced the trustworthiness and rigor of the study.

### Ethics declarations

This study was approved by Ethics Committee of Yibin Second People's Hospital (#2024-054-01).

### Inclusivity in global research

Additional information regarding the ethical, cultural, and scientific considerations specific to inclusivity in global research is included in the Supporting Information (SX Checklist).

## Results

Twenty-six interviews (13 young breast cancer patients and 13 spouses) of the participants were conducted and included in the analysis of this study. Table 1 presents the demographic characteristics of the participants. The analysis of interview data identified two main themes, each comprising three to four subthemes: dyadic coparenting through constructive communication: a shared journey of navigating parenting challenges, which includes open communication of parenting experiences, emotional co-regulation in parenting, and positive communication of parenting plans; and negative co-parenting communication: protect parenting emotions or not provide emotional support, which includes "instrumental support" and emotional avoidance, avoid communication to prevent emotional fluctuations, communication concealment to avoid increasing parenting pressure, and invalid communication without providing parenting emotional support.

### Theme 1. Dyadic coparenting through constructive communication: A shared journey of navigating parenting challenges

It refers to couples acting as a coping dyad, gradually developing shared parenting strategies (such as joint decision-making, role complementarity, and emotional coordination) through constructive communication when facing the parenting challenges brought by illness [37].

**Sub-theme 1. Open communication of parenting experiences.** The sub-theme reflects a pattern where parents share their challenges, concerns, and strategies regarding parenting openly with each other. This openness appears crucial for navigating the difficulties they face and for enhancing mutual support. Three young breast cancer patients and their spouses were able to communicate their experience of parenting underage children after illness.

**Table 1. Demographics of participants (N = 26, P = 13, S = 13).**

| Patient ID | Age of patient（year） | Number of children | Age of the youngest child (years) | Stage of cancer | Education background of patient | Occupation of patient | Residence | Spouse ID | Age of spouse（year） | Occupation of spouse | Education background of spouse |
|---|---|---|---|---|---|---|---|---|---|---|---|
| P1 | 38 | 1 | 7 | Period I | College | Staff | City | S1 | 41 | Staff | College |
| P2 | 42 | 2 | 11 | Period II | Middle school | Farmer | Town | S2 | 47 | Farmer | Middle school |
| P3 | 38 | 2 | 2 | Period III | Postgraduate | Staff | City | S3 | 40 | Staff | Postgraduate |
| P4 | 39 | 1 | 4 | Period II | College | Staff | City | S4 | 41 | Staff | College |
| P5 | 33 | 2 | 2 | Period I | Primary school | Farmer | Town | S5 | 34 | Farmer | Primary school |
| P6 | 35 | 2 | 6 | Period I | College | Government staff | City | S6 | 34 | Staff | College |
| P7 | 44 | 2 | 11 | Period III | Primary school | Farmer | Town | S7 | 50 | Farmer | Primary school |
| P8 | 38 | 1 | 6 | Period II | Undergraduate | Staff | City | S8 | 38 | Staff | Postgraduate |
| P9 | 43 | 3 | 14 | Period III | Primary school | Farmer | Town | S9 | 50 | Farmer | Primary school |
| P10 | 34 | 1 | <1 | Period I | Undergraduate | Staff | City | S10 | 33 | Staff | Undergraduate |
| P11 | 35 | 2 | 3 | Period III | College | Staff | City | S11 | 40 | Staff | College |
| P12 | 41 | 2 | 7 | Period II | Primary school | Farmer | Town | S12 | 40 | Farmer | Primary school |
| P13 | 36 | 1 | 3 | Period II | Postgraduate | Staff | City | S13 | 40 | Staff | Postgraduate |

Note: P = Patient, S = Spouse.

*I haven't been with my child for a long time, I'm not very familiar with his habits. I have to ask my wife about many things, and I exchange experiences and ideas with her. (S11, 40 years, staff)*

*He's not quite sure about the child's habits, I told him. (P11, 35 years, 2 children, Period III)*

*My daughter is 3 years old. After I fell ill, my husband took the initiative to take care of my daughter. We discussed our experiences in taking care of my daughter and exchanged experiences with each other. (P13, 36 years, 1 child, Period II)*

*I discuss my daughter's matter with her. (S13, 40 years, staff)*

**Sub-theme 2. Emotional co-regulation in parenting.** Emotional Co-Regulation in Parenting involves intentional efforts by parents to manage and support each other through emotional challenges related to parenting and personal hardships. This proactive approach includes offering comfort, addressing concerns directly, and collaboratively finding solutions to reduce stress and anxiety. Four couples understand and tolerate each other's emotions, and will alleviate negative emotions through communication.

*When I think of my two children at night, I can't help but shed tears. What should they do when I'm gone! My husband comforted me and advised me not to overthink. He said that my illness will get better and my children will grow up. (P2, 42 years, 2 children, Period II)*

*Since she was diagnosed, she has been worried about various things and she asked me if she's gone, will I do my best to raise the child. I told her that the medical technology is already mature now, and there are also many people suffering from this disease, which is just a common illness. I tried to alleviate her concerns. (S2, 47 years, farmer)*

**Sub-theme 3. Positive communication of parenting plans.** Positive Communication of Parenting Plans involves collaboratively developing and discussing strategies for effective parenting in the face of challenges. This approach emphasizes constructive and forward-looking dialogue about how to manage parenting responsibilities, ensuring that plans are made in a supportive and solution-oriented manner. 4 couples actively communicate about parenting plans for their underage children.

*I have two children, the oldest is 7 years old and the youngest is 2 years old. (I work out of town) So my wife takes care of the children alone. It's not possible now. My wife and I have discussed that my parents in law will take care of the children. (S3, 40 years, staff)*

*We discussed seeking help from my parents to take care of the child. (P3, 38 years, 2 children, Period II)*

*My eldest daughter has dropped out of school, and I am worried that she will endure hardship. However, I don't have much energy to guide her because of my illness!"(sigh). My son can't do this anymore. My husband and I have discussed we will arrange for the child to attend primary school in a county-level school. After the chemotherapy is over, I will accompany him to study. (P7, 44 yaers, 2 children, Period III)*

*We have decided to arrange our son to a county-level school. (S7, 50 years, farmer)*

**Theme 2. Negative co-parenting communication: Protect parenting emotions or not provide emotional support**

Negative co-parenting communication refers to the occurrence of negative emotions during the communication process, it usually manifests as criticism, blame, indifference, or avoidance of communication [38].

**Sub-theme 1. "Instrumental support" and emotional avoidance.** This sub-theme describes a dyadic coping pattern in which one partner (typically the husband) provides practical, task-oriented support (instrumental support) while intentionally or unintentionally avoiding emotional discussions related to parenting challenges.

Spouse (6/13) tends to provide practical problem-solving (such as contacting school, arranging for the transportation of children to school), but avoids emotional communication.

*I don't know how to comfort people. It's my responsibility to make money and take care of children. Leaving the crying to her. (S4, 41 years, staff)*

*He only does things silently, but never talks to me about my anxiety about children.(P4, 39 years, 1 child, Period II)*

*Last month, I felt physically exhausted and emotionally overwhelming after chemotherapy. I said to him, "Our child needs help with math homework, but I truly have no energy left to assist." When I said this, I hoped he would notice my emotion first. I wished he would sit down, even if just to hold my hand. But he didn't. After hearing me, he didn't say a word, just immediately stood up and walked into our child's room.(P9, 43 years, 3 children, Period III)*

*Fixing the problem is what I know how to do. (S9, 50 years,farmer)*

*Once when our child had a fever, I completely broke down, sobbing and blaming myself, I told him "Is this because I passed on my illness?" He cut me off, saying, "This isn't the time for crying," and then he took the child to the hospital, leaving me alone at home to cry.(P10, 34 years, 1 child, Period I)*

*Some hurdles can't be overcome just by talking. As long as I keep this household steady, with time, the wounds will eventually heal. (S10, 33 years, staff)*

This resulted in five wives believe their husbands are "emotionally indifferent", while five husbands felt powerless.

*He is very indifferent to me and never cares about my feelings. (P11, 35 years, 2 children, Period III)*

*There's nothing to say. I feel powerless. (S11, 40 years, staff)*

*I have become a burden. He only cares about the tasks at hand and never pays attention to how I'm feeling physically or emotionally.(P12, 41 years, 2 children, Period II)*

*I don't know how to offer emotional reassurance. To me, resolving practical problems is the most substantial form of support.(S12, 40 years, farmer)*

**Sub-theme 2. Avoid communication to prevent emotional fluctuations.** Avoiding Communication involves withholding personal concerns and emotional struggles from one's partner to prevent negative emotions. This pattern is evident when an individual, despite having significant worries about their children's well-being and the impact of their illness, chooses not to share these concerns with their spouses. The intent is to avoid causing negative emotions, which could result in the individual enduring their discomfort alone. This reluctance to communicate reflects a protective measure to protect oneself from additional stress, but may also result in unresolved emotional strain for the individual. Some patients choose to avoid discussing parenting related topics in order to avoid causing emotional fluctuations.

*I haven't told him my concerns, I don't want to (cry), we have different opinions, and we argue as soon as we talk. Whenever we argue, it affects my emotions, and I don't want to suffer from emotional pain. (P4, 39 years, 1 child, Period II)*

*I don't want to talk about with her. (S4, 41 years, staff)*

**Sub-theme 3. Communication concealment to avoid increasing parenting pressure.** Communication Concealment is characterized by the deliberate withholding of concerns, pressures, and emotional struggles from one's partner to shield them from additional stress. This approach is evident when individuals choose not to share their worries about their child's potential difficulties or the financial strain related to illness, despite these issues being significant. For example, one spouse may refrain from discussing fears about their child's experience at school or the impact of financial burdens on their partner's health, opting instead to manage these concerns privately. This concealment is motivated by a desire to prevent exacerbating the partner's psychological burden, but it can result in increased individual stress and lack of mutual support. Two young breast cancer patients chose to conceal communication with their spouses about their concerns on children.

*I am worried that my child will be mocked by his classmate for my illness. But I don't tell my husband and my child (with a bitter smile). I'm almost depressed now and I feel like I can't get out! (P2, 42 years, 2 children, Period II)*

*My wife is sick and I can't put too much pressure on her anymore. I just need to endure the pressure alone. I don't tell her (my pressure), usually only say good things. (S2, 47 years, farmer)*

**Sub-theme 4. Invalid communication without providing parenting emotional support.** Invalid Communication is characterized by responses that dismiss, ignore, or inadequately address one partner's expressed concerns, resulting in feelings of invalidation and lack of support. This is evident when one partner's worries about how their illness might impact their child's education are met with dismissive comments, such as being told they are overreacting (P1). Similarly, when concerns about the child's well-being are shared but met with disregard, such as the partner focusing on their phone instead of engaging in the conversation (P4), it highlights a failure to provide meaningful support. This form of communication undermines the emotional needs of the concerned partner and contributes to a breakdown in effective and empathetic dialogue. Two interviewees were unable to receive corresponding emotional support from their spouses when they expressed their concerns and parenting ideas about their underage children to their spouses.

> I told him I was worried that my illness would affect my child's learning. (P1, 38 years, 1 child, Period II)
>
> I think she was worrying too much. (S1, 41 years, staff)
>
> I told him about my concerns about the child, but he only looked at his phone and ignored me. (P4, 39 years, 1 child, Period II)
>
> I don't want to talk about the topic with her. (S4, 41 years, staff)

## Discussion

This study aimed to explore the communication experience in parenting underage children among young breast cancer couples from the dyadic perspective. In this study, we have interviewed 13 couples with breast cancer. Actually, after interviewing 13 couples, considerable data were repeated in the present study, which meant that data saturation had been reached. Therefore, comprehensive information regarding the communication experience in parenting underage children among young breast cancer couples was obtained in this study. The findings provide deep insights into the communication experience in parenting underage children among Chinese young breast cancer patients and their spouses, which may guide the development of culturally tailored interventions to improve the communication methods between couples related to parenting issues, and to enhance their ability to make joint decisions, and to reduce their negative emotions.

Our study found that most young breast cancer patients and their spouses avoid discussing topics of parenting underage children, and only four couples actively communicated about their experience of raising underage children, which may be related to their avoidance of parenting stressors. This is consistent with Reblin's findings [39]. Communication between patients and spouses is an important resource for managing cancer needs [40]. Research has shown that communicating and sharing emotions between cancer patients and their spouses can help improve their self-efficacy and intimacy satisfaction, as well as improve their mental health [41]. Study revealed that young breast cancer patients will be passively reduced from the task of raising children due to cancer treatment [42]. As the main supporter of the patient, the spouse will not only take care of the patient but also take care of underage children. Both the patient and spouse will change their original family roles, which is likely to lead to problems of poor family adaptation [42]. Therefore, enhancing the communication and parenting experience between couples can promote the improvement of their self-efficacy and mental health. During the treatment period, breast cancer patients may face insufficient ability to raise underage children and uncertainty, which may change the beliefs and expectations of couples in parenting [43]. Parents are willing to bear the pressure for their children's growth. Whenever there is a major change (such as a cancer diagnosis), they do not want the event to affect their children's emotions, daily life, or education. Therefore, parents communicate with each other to facilitate a smooth transition of childcare responsibilities without negatively impacting their children [11]. Through communication, cancer patients and their spouses can jointly explore their values, educational perspectives, and plans regarding child parenting, which can alleviate patients' anxiety and enhance their spouse's support [18]. There is an intervention program in

a western country that focuses on communication between couples [44], and the results show that improving the level of communication between couples can alleviate anxiety and depression in both partners. This finding suggests that healthcare providers should fully leverage the important role of communication and parenting experience between young breast cancer patients and their spouses, and develop targeted intervention strategies to improve family function.

Our findings reveals a significant pattern in how young Chinese breast cancer couples negotiate parenting responsibilities while managing breast cancer, characterized by an emphasis on practical support coupled with emotional disengagement. In this study, a dyadic coping pattern in which one partner (typically the husband) provides practical, task-oriented support (instrumental support) while intentionally or unintentionally avoiding emotional discussions related to parenting challenges. Studies have shown that gendered expectations can influence how individuals cope with illness and how they express their emotions [45]. Specifically, women are often expected to express their emotions more freely than men, who may be socialized to suppress their emotions and appear strong and stoic in the face of adversity.

Findings highlighted in a recent systematic review demonstrate that limited disclosure of both emotions and thoughts around cancer-related concerns within a couple may compromise relational and psychological wellbeing of both members of the couple, including higher symptoms of depression and anxiety (i.e., negative affect) or lower symptoms of wellbeing and closeness (i.e., positive affect) [46,47]. When patients are unable to fulfill their parenting responsibilities due to treatment, communication between couples may be highly "transactional" (such as only discussing picking up and dropping off children, homework guidance), ignoring emotional support needs [48]. Husbands in our study typically demonstrated their care through concrete actions, such as take care of children (S4: "I don't know how to comfort people. It's my responsibility to make money and take care of children."), work silently (P4: "He only does things silently, but never talks to me about my anxiety about children."). The Chinese case reveals that neglecting the culturally embedded definitions of "support" and "care" makes it difficult to grasp the underlying logic of cancer patients' intimate interactions. Therefore, global research on cancer communication urgently needs to establish a cultural interpretative framework—one that acknowledges universal stressors while placing greater emphasis on how specific cultures construct the meanings of illness, gender roles, and expressions of intimacy.

Communication concealment was another important theme identified in this study. In order not to increase the anxiety of the spouse, two young breast cancer patients chose to conceal communication with their spouses about their concerns about children. The observed phenomenon reflects a complex interplay of parental protective silence and family system maintenance. Existing literature suggests this behavior serves as a dyadic coping strategy to shield partners from the guilt of parental role disruption [49] while preserving family boundaries [50]. Breast cancer treatment will damage the patient's physical function, and patients often worry about the adverse effects on underage children, especially school-age children. Parents want to show their best side in front of their children's friends and classmates, fearing that their image and physical condition may affect others' perception of their children. When patients cannot balance the roles of parents and patients, and cannot fulfill their parenting responsibilities well, they will be more concerned about the impact of their illness on their children. Accordingly, some of our participants stated that they were concerned that children would be mocked by their classmates (P1: "I told him I was worried that my illness would affect my child's learning."). It can be seen that young patients with breast cancer are prone to psychological distress. In addition, patients cannot take care of their underage children as they did before illness, the pressure to take care of children is mostly on their spouses, who often face enormous psychological and economic pressure. In order to avoid causing additional burden on their spouses, some patients choose to conceal their inner thoughts about raising underage children. This is consistent with Manne's findings [15]. This behavior is particularly prevalent in collectivist cultures, such as those in East Asia, where family harmony and collective well-being are prioritized over individual expression [51]. Patients often believe that withholding their worries can protect their spouses from anxiety and maintain a stable environment for their children [52].

However, while such Concealment may temporarily reduce relational tension [53], longitudinal studies indicate it can exacerbate emotional disconnection and impair co-parenting coordination [54]. Previous studies have shown that

concealed communication is a behavior of actively suppressing and consuming emotional and cognitive resources, which can lead to increased psychological stress and a certain degree of anxiety and depression in patients [55], and the ultimate effect is often opposite to patients' expectations [39]. There is a research confirming that timely communication between cancer patients and their spouses can reduce their emotional distress, improve their intimacy and relationship satisfaction, and enhance their disease adaptation ability [56]. Research has shown that cancer patients and their spouses can alleviate the stress and negative emotions through open communication [57]. Clinically, interventions promoting measured openness (e.g., structured future-planning dialogues may help couples balance protection with authentic communication [58]. Healthcare providers need to promote open communication related to parenting among couples of young breast cancer patients and encourage emotional expression and response through empathy or self-disclosure interventions. Additionally, it is also worth noting that the participants in this study reported economic burden was also one of the problems they concealed in communication, as it can bring psychological pressure to both spouses. It is suggested that healthcare professionals should pay more attention to young breast cancer patients with high economic burdens, and recommend appropriate treatment schemes for them.

Avoidance of communication and Invalid communication was the other negative communication methods mentioned by the participants. In the case of cancer, due to anxiety and helplessness caused by concerns about underage children, patients need emotional support from their spouses more. When their spouses do not respond or respond in an argumentative manner, patients may experience various negative emotions. In this study, most spouses are unwilling to communicate positively with patients about issues related to raising underage children, and patients are unable to express their inner emotions, leading to increased psychological distress. This finding is similar to that of previous studies [22]. Firstly, the reason may be that cancer diagnosis disrupts the emotional and physical intimacy between spouses. However, the physical and psychological toll of cancer can create emotional distance or role shifts, weakening the bond between couples [59]. Couples mistakenly believe that talking will only make things worse, previous study indicated that cultural expectations (like the "strong parent" ideal in Asian families) make it even harder to break [60]. Second, for young breast cancer patients and their spouses, the added stress of parenting responsibilities exacerbates communication challenges. Patients may suppress their emotional needs out of guilt for not fulfilling their parental roles, while spouses may feel overwhelmed by the increased caregiving burden. This dynamic creates additional tension and hinders effective communication. However, this lack of communication often exacerbates the psychological burden on patients, creating a cycle of emotional isolation and distress [61]. Finally, young breast cancer patients and their spouses lack empathy and communication skills, and the spouse is not aware of the importance of spousal communication for the patient's mental health [62,63]. Research has shown that guiding spouses to listen, ask questions, and empathize with patients through case analysis, skill practice, role-playing, and homework can help couples to effectively communicate [64]. Therefore, healthcare providers need to help young breast cancer couples seek communication skills to improve their communication methods, and reduce ineffective communication. However, based on the above possible reasons, in addition to improving the communication skills of patients and their spouses, it is also worth considering providing formal (such as economic and psychological assistance from social workers, treatment subsidies from public welfare foundations, childcare services provided by the government or enterprises) or informal (such as mutual assistance between families and neighbors support systems, sharing experiences with patients in the community) supports through multi-party collaboration such as medical- community-public welfare organizations, which is expected to further alleviate the parenting burden and psychological pressure of patients and their spouses.

## Practical implications

The findings in this study with regard to couples' communication are relevant for cancer psychological care, cancer support policies, and society, as an increase in cancer patients raising underage children will lead to a greater demand for cancer support. With regard to the fact that young couples with breast cancer are both open communication of coping

parenting difficulties and negative communication for protecting parenting emotions or not provide emotional support, we propose that in clinical practice, healthcare professionals could develop tiered intervention programme based on differences in communication patterns: for couples inclined toward open communication, their skills in shared decision-making and emotional co-regulation should be enhanced; for couples exhibiting protective silence, safe spaces for emotional expression should be facilitated through structured communication training. Furthermore, it is recommended that policymakers incorporate a family communication support module into existing cancer care guidelines to help families enhance their coping resilience.

## Limitations

Limitations for this study should be considered. This study was performed with Chinese participants, which is why certain aspects cannot reflect the parenting-related communication experiences of young breast cancer patients and their spouses from other countries, given the particularities of Chinese culture. Likewise, As a qualitative interview study, findings rely on participants' subjective recollections and interpretations, which may be influenced by social desirability or recall bias. In addition, while thematic saturation was achieved for major themes, some nuanced sub-themes might benefit from further exploration with larger samples. Despite these limitations, we hope that our findings can contribute to better understanding the experiences of young breast cancer patients who parenting underage children.

## Conclusion

This study contributes to an in-depth understanding of the communication experiences of Chinese young couples with breast cancer as they navigate the challenges of raising underage children. The findings of this study provide valuable suggestions for patients and healthcare professionals regarding parenting issues among young couples with cancer.

In this study, various communication methods used by young breast cancer couples dealing with parenting issues were identified, including open communication of parenting experiences, emotional co-regulation in parenting, positive communication of parenting plans, "instrumental support" and emotional avoidance, avoid communication to prevent emotional fluctuations, communication concealment to avoid increasing parenting pressure, and invalid communication without providing parenting emotional support. Given that communication styles are influenced by cultural factors and individual skills, healthcare professionals should design culturally sensitive interventions. These interventions should aim to improve communication awareness and engagement among young breast cancer couples, particularly in the context of parenting young children.

## Supporting information

**S1 File. Inclusivity-in-global-research-questionnaire.**
(DOCX)

**S1 Text. Interview transcript text.**
(DOCX)

## Acknowledgments

All authors were grateful to the participants for their participation and sharing.

## Author contributions

**Conceptualization:** yingchun Li, Meichan Chong, Pinglei Chui.

**Data curation:** yingchun Li, Yuman Yuan, Qiaoli Zhong.

**Formal analysis:** yingchun Li, Qiaoli Zhong, Maoting Tang.

**Methodology:** Maoting Tang, Yanjia Liu.

**Supervision:** Meichan Chong, Lin Mo, Liande Tao.

**Writing – original draft:** yingchun Li.

**Writing – review & editing:** yingchun Li, Meichan Chong, Pinglei Chui, Lin Mo, Liande Tao, Yanjia Liu.

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
